# Bringing Parent–Child Interaction Therapy to South Africa: Barriers and Facilitators and Overall Feasibility—First Steps to Implementation

**DOI:** 10.3390/ijerph19084450

**Published:** 2022-04-07

**Authors:** John-Joe Dawson-Squibb, Eugene Lee Davids, Rhea Chase, Eve Puffer, Justin D. M. Rasmussen, Lauren Franz, Petrus J. de Vries

**Affiliations:** 1Division of Child and Adolescent Psychiatry, Department of Psychiatry & Mental Health, Faculty of Health Sciences, University of Cape Town, Cape Town 7700, South Africa; eugene.davids@uct.ac.za (E.L.D.); lauren.franz@duke.edu (L.F.); petrus.devries@uct.ac.za (P.J.d.V.); 2Judge Baker Children’s Center, Harvard Medical School, Boston, MA 02115, USA; rchase@jbcc.harvard.edu; 3Department of Psychology & Neuroscience, Duke Global Health Institute, Duke University, Durham, NC 27708, USA; eve.puffer@duke.edu (E.P.); justin.rasmussen@duke.edu (J.D.M.R.); 4Division of Child and Adolescent Psychiatry, Department of Psychiatry and Behavioural Sciences, Duke Global Health Institute, Duke University, Durham, NC 27708, USA

**Keywords:** implementation, low- and middle-income countries, child and adolescent mental health, Parent–Child Interaction Therapy

## Abstract

There is a large assessment and treatment gap in child and adolescent mental health services, prominently so in low- and middle-income countries, where 90% of the world’s children live. There is an urgent need to find evidence-based interventions that can be implemented successfully in these low-resource contexts. This pre-pilot study aimed to explore the barriers and facilitators to implementation as well as overall feasibility of Parent–Child Interaction Therapy (PCIT) in South Africa. A reflective and consensus building workshop was used to gather South African PCIT therapist (N = 4) perspectives on barriers, facilitators, and next steps to implementation in that country. Caregiver participants (N = 7) receiving the intervention in South Africa for the first time were also recruited to gather information on overall feasibility. Facilitators for implementation, including its strong evidence base, manualisation, and training model were described. Barriers relating to sustainability and scalability were highlighted. Largely positive views on acceptability from caregiver participants also indicated the promise of PCIT as an intervention in South Africa. Pilot data on the efficacy of the treatment for participating families are a next step. These initial results are positive, though research on how implementation factors contribute to the longer-term successful dissemination of PCIT in complex, heterogeneous low-resource settings is required.

## 1. Introduction

Globally, there are challenges and on-going efforts to improve access to evidence-based and evidence-informed psychological and psychosocial treatments [1,2]. The mental health of children and adolescents in particular is regarded as a global priority [3]. Children and adolescents represent approximately 44% of the world population and about 90% of them live in low- and middle-income countries (LMICs) [4,5]. There is a large assessment and treatment gap in Child and Adolescent Mental Health services (CAMHS), prominently so in low- and middle-income countries [6,7,8,9,10]. The impact of these gaps includes limited access to early detection, screening, and delays in access to early intervention for mental health challenges [11]. In South Africa, for example, a recent review by Mokitimi et al. [10] indicated that potentially fewer than 10% of the expected population of children and adolescents in need of mental health care received clinical care provision. The review further described the very limited access to psychosocial treatments at all levels of mental health services [10]. The divide between the need for and access to CAMH services in South Africa was summed up by Flisher and colleagues [9] who wrote, “the relatively inaccessible and underdeveloped mental health services, in the face of considerable need for services, begs the question of where children and adolescents with mental health problems receive help” (p. 157).

One of the most common groups of mental health conditions is disruptive behavioural difficulties (DBD) which are a significant concern worldwide. Estimates in South Africa, for example, are that up to 6% of children have Oppositional Defiant Disorder [12]. Untreated or undertreated child behaviour problems are frustrating and demoralizing for children and families and costly to society, at times contributing to a cascade of health and social problems affecting children’s long-term health, well-being, and prospects for economic self-sufficiency [13]. Young children with such disorders are more likely than their peers to experience emotional and physical abuse, academic difficulties, peer rejection, and other mental health difficulties, including depression and anxiety [14,15,16]. In addition, later in their lives these children are more likely to engage in physical abuse, delinquency, substance abuse, and suffer from other mental health problems, including, depression and personality disorders [17,18]. These conditions often present at community mental health centres, putting additional burden on the system [19].

Given the treatment gap in CAMH and the high rate of externalising behaviours in the country there is a pressing need to find evidence-based interventions to meet the significant need. In the context of ASD and neurodevelopmental disabilities, de Vries [11] recommended that LMIC should establish a “pyramid of interventions” across primary, secondary and tertiary levels of care that can meet the needs of children and families in a step-wise manner. The same principle would be applicable to other CAMH disorders.

### 1.1. Parent–Child Interaction Therapy

Parent–Child Interaction Therapy (PCIT) is one intervention that addresses disruptive behaviours in children and has a large evidence base in a range of diagnostic populations and at primary, secondary and tertiary service levels [20,21,22,23,24,25,26,27]. PCIT is considered one of the most efficacious parent management training programmes available for young children with disruptive behaviour problems [20]. PCIT leads to significant decreases in child externalizing behaviours and increased child compliance [28,29,30,31,32]. Research also describes improvements in multiple child and caregiver outcomes, including parenting stress, caregiver–child attachment, and child internalizing symptoms [28,33,34].

Based in attachment and behaviour theories, PCIT differs from many other parent training programs in a number of key ways. Rather than a primarily didactic approach, therapists coach parents in real time with their child, typically using a one-way mirror and earpiece [20]. Parents are taught relationship enhancement and discipline skills that are then practiced both at home and in weekly, individual sessions. Parents are also expected to meet certain goals in order to progress through the model and then to graduate from the programme. PCIT is administered in a manualised format, and is specifically designed for families with children between ages 2 and 7 years old. Its manualised structure and emphasis on fidelity to the protocol is regarded as a strength and facilitates the intervention to be scaled up and disseminated and still maintain efficacy. While PCIT has been shown to be cost effective, its treatment duration, clinician training requirements, and delivery models have led some authors to note the implications of these in relation to effectiveness, cost and acceptability for parents [27,35]. There is, therefore, a need to assess the feasibility of PCIT in different contexts.

There is some evidence of feasibility research on PCIT [36,37], e.g., treatment barriers, participation, and fidelity to the model. Further, some studies have shown that adaptation to the standardised PCIT programme have allowed for accessibility in diverse settings and with a range of cultural groups [38,39,40]. Abrahamse et al. [41] posit that the inherent flexibility of PCIT allows for sensitivity and responsiveness to cultural variations, ensuring that it can be readily implemented in international settings.

As described by Lieneman et al. [31] in their review of PCIT research between 2006–2017, there is evidence of PCIT being examined in a range of contexts and countries. Apart from the USA, where the majority of research has taken place to date, there is evidence from other countries, including the Netherlands [42], Norway [43], Hong Kong [44], Australia [45], and New Zealand [46]. There is further evidence for efficacy of the intervention for families from a range of racially and ethnically diverse backgrounds, including Puerto Rican, Mexican-American, Alaskan native, and Chinese families [47]. However, the current evidence base for PCIT is drawn exclusively from high-income countries (HIC) [31].

The evidence above suggests that PCIT can be delivered successfully in a range of contexts. While this indicates that PCIT may be a suitable intervention in LMIC, it requires thorough evaluation. There is therefore a critical need in the literature relating to the feasibility, effectiveness and acceptability of PCIT in LMIC, where cultural, linguistic, and socio-economic diversity and disparities are likely to be significant. Implementation evaluations in such contexts would prove invaluable.

### 1.2. Implementation Science: Bringing Evidence-Based Intervention to LMIC

To integrate evidence-based interventions into multicultural global settings, implementation context must be considered [48]. Damschroder et al. [49] describe implementation science as a method of enquiry designed to support investigators in determining whether interventions or approaches can be implemented in real-world settings. They suggest that these settings may differ from the original setting in many ways [49]. Bammer [50] reflects that implementation science can also facilitate bridging the gap between research and practice. It further enhances an appreciation of how to improve collaborative processes in research, e.g., ensuring that appropriate researchers and sectoral representatives are included, and their interests are accommodated. Implementation outcomes as proposed by Proctor et al. [51] can include factors including acceptability, appropriateness, feasibility, and sustainability. Table 1 below provides relevant definitions of key implementation terms. Several studies have considered implementation and PCIT though have primarily focused on provider training, as well as trainee and organisational factors [31,52]. An awareness of cultural diversity and contextual factors including resource limitations is particularly salient to LMIC.

The Consolidated Framework for Implementation Research (CFIR) is a comprehensive and popular framework that presents a taxonomy for distinguishing between a range of contextual determinants of implementation success [49]. A recent study by Means et al. [56] proposed an optimized version of the model for use in LMIC. The optimised framework proposes additional domains and constructs to the original to make it more relevant to LMIC contexts, for example, perceived scalability, sustainability, and external funding agents. The CFIR model can be used as a pragmatic framework to explore implementation, including during the pilot testing phase [56]. The authors conclude that further study is required to determine if their proposed additions to the original CFIR model are reliable and valid [56]. Figure 1 below represents the CFIR and the adapted six domains for LMIC proposed by Means and colleagues.

Taking together the high rates of DBD and intervention gaps in LMIC, and the limited evidence base in these low-resource, highly diverse contexts, various authors have highlighted the benefits of a pragmatic evaluative approach that focuses on feasibility and acceptability during pilot testing phase of intervention studies [48,57,58]. As an example, Makombe et al. [59] demonstrated in their feasibility evaluation of an early intervention for autism spectrum disorder in South Africa, that implementation factors like complexity of intervention, logistical constraints (e.g., time, Internet access), and mismatch between programme content and the local context were all barriers to successful scalability of the programme. Literature in this area has also identified the importance of including stakeholders in feasibility and implementation research [10,48]. Such stakeholders are valuable in providing a good understanding of the local context and setting which facilitates better and more successful implementation [48].

PCIT has been identified by clinicians and researchers in a joint University of Cape Town and Duke University project as a possible intervention for use in South Africa. However, little is known about the efficacy and implementation of the intervention in a community setting in LMIC. A larger study aims to explore the feasibility, acceptability, and preliminary indicators of efficacy of PCIT in a South African setting, using an implementation science approach. South Africa is a diverse country of around 61 million people, with eleven official national languages [60]. This current study had a primary aim to explore the barriers and facilitators to implementation, as well as overall feasibility to determine best next steps for PCIT in South Africa. As a secondary aim it sought to explore the acceptability of the intervention for caregivers. Using two key stakeholder groups, caregivers and therapists, it sought to provide early feasibility evidence for PCIT in this diverse, low-resource setting.

## 2. Methods

### 2.1. Design

The study primarily employed a qualitative descriptive approach to explore the barriers and facilitators to implementation as well as elements of overall feasibility to determine next steps for the potential delivery of PCIT in South Africa. In addition, it gathered limited quantitative data relating to caregiver characteristics, treatment satisfaction and therapist fidelity.

### 2.2. Participants

#### 2.2.1. Clinicians

A child and adolescent mental health clinic based at a tertiary level Children’s Hospital in Cape Town, South Africa enrolled clinicians in training in PCIT. Four (three male and one female) Clinical Psychologists were selected through convenience sampling and invited to participate in the study. The clinical experience of these clinicians ranged from 15 to 41 years.

#### 2.2.2. Caregivers

The same clinic enrolled seven caregiver dyads, representing 14 caregiver participants in the study. The seven dyads each had a child in their care, ranging in age from 2½ years to 7 years.

### 2.3. Contextual Background and Procedures

The current study formed part of a broader research project that aimed to evaluate the feasibility, acceptability and preliminary indicators of efficacy for PCIT in a South African setting, using an implementation science approach. That project came through an identified area of mutual interest between the University of Cape Town and Duke University. Following discussions and a collaborative meeting, PCIT was selected as a priority area. Pilot funding from Duke Global Health Initiative Cape Town priority partnership supported the training and supervision of the clinicians. As part of that effort, four South African Clinical Psychologists completed an initial 40-h PCIT training by a certified PCIT Global Trainer at a child and adolescent mental health clinic, in Western Cape, Cape Town. The Western Cape is a diverse province of South Africa, with 3 high-frequency languages (English, Afrikaans, and Xhosa) and many different cultures. The clinical head of the child and adolescent mental health clinic was included during the preparatory stages of the study and permission was sought to engage with the clinicians. Participating clinicians were provided with information about the study, and were asked to provide written, informed consent. Following the initial 40-h input, they continued the standard training process under regular online supervision from the Global trainer, as they worked toward formal PCIT certification during the course of this research study. This included seeing PCIT cases at the clinic, following the standard manual protocol and providing the supervisor with recordings of their sessions, allowing regular and direct feedback of their work. The clinicians kept reflective notes during the process, describing cultural and other barriers and facilitators relating to the implementation of PCIT in South Africa. The seven cases were selected using convenience sampling, and had been referred to the clinic for treatment. In some cases, they were specifically referred to the clinic with the intention of receiving PCIT as part of this study. They were not selected to be representative of the diverse South African population. Following consent from the seven families undergoing PCIT treatment in this study, data were collected from these cases, including pre, during and post-intervention. On completion of the cases, participating clinicians were invited to a stakeholder reflection and consensus-building workshop.

### 2.4. Data Generation

Data were generated using a stakeholder reflection and consensus-building workshop approach. The facilitator of the workshop (who was also one of the PCIT clinicians and a participant in the study), generated a list of questions related to the clinicians’ experiences of using PCIT in their clinical practice, and examined the perceived barriers and facilitators to implementing PCIT. In addition, the clinicians were asked to reflect on the acceptability of the therapeutic modality in the community and context in which they work. The regular reflective diaries kept during the course of the work was also reviewed prior to the workshop. Using the suggestion by Peterson and Barron [61], sticky notes were used to stimulate discussion and get participants to reflect on some of the questions asked. After the clinicians reflected on the questions posed, and wrote one point per sticky note they then started pasting it on flipchart paper. The notes were then grouped together by participants based on their similarity in response. After all the participants wrote down their reflections about the barriers and facilitators of PCIT in the local setting as well as the acceptability of this innovative therapeutic modality in the South African setting, the facilitator led the focus group discussion among the participants by referring to the points on the sticky notes. The consensus-building workshop was audio-recorded and was informed by the interview schedules guiding questions as well as the responses on the sticky notes. Participants were then able to reflect upon the points which they highlighted on the sticky notes and allowed for a deeper and richer description of the barriers, facilitators and thoughts on acceptability of PCIT. The use of the sticky notes followed by the consensus-building workshop also allowed for member checking and enabled the facilitator to ask clarifying questions to ensure that the reflections were captured and understood correctly.

Though still under supervision and in the process of becoming certified, therapist fidelity data were gathered following each PCIT session, and therapists were recorded to what extent they had kept to the treatment manual. Caregivers were also asked to complete the Therapeutic Attitude Inventory and the four treatment satisfaction questions on completion of the treatment programme. Figure 2, below, details the flow of research.

### 2.5. Measures

Therapeutic Attitude Inventory and treatment satisfaction questions (Brestan et al. [62])

The TAI is a 10-item measure used to establish parents’ satisfaction with therapy [62]. Parents rate each of the items on a scale of 1 to 5, with 5 indicating greater satisfaction and 1 the opposite [25]. Very basic treatment satisfaction questions developed for this study were also asked.

Therapist’s self-assessment of fidelity (Funderburk and Eyberg [63])

The PCIT treatment protocol [63] includes integrity checklists for each session which list the key therapeutic activities considered integral to the conduct of PCIT. Participating therapists were asked to complete an integrity checklist after the completion of each session.

### 2.6. Data Analysis

After the workshop, the audio-recording was transcribed by an independent research assistant. In addition, all sticky notes were typed up into one document. The transcription together with the document containing all the points on the sticky notes were analysed by two researchers (JJDS, ELD), first independently, and then together to reach consensus on findings.

The data were analysed using framework analysis. Framework analysis affords researchers the freedom and flexibility to analyse data in a way that is best suited for the particular aim and is best suited to research questions that seek to contextualise and evaluate interventions [64,65]. Framework analysis forms part of the broad family of analysis methods such a thematic and content analysis, where it attempts to identify similarities and differences in qualitative data [66]. Framework analysis is made up of 6 steps, namely: (i) transcription of the interview; (ii) familiarisation of the interview; (iii) coding of the interview; (iv) then to develop a working analytical framework; (v) followed by applying the analytical framework, in the case of the current study the data were coded then examined using the Consolidated Framework for Implementation Research (CFIR); and, finally, (vi) charting data into a framework matrix [66]. The CFIR provides both researchers and clinicians with a framework for understanding effective implementation of interventions from the perspective of implementation science. The framework is well-suited to guide evaluations of complex health care delivery interventions, as it provides a comprehensive model to systematically identify factors that can emerge in multi-level contexts which influence implementation [67]. As a pragmatic tool, the CFIR model can be used to explore implementation, including during the pilot testing phase, and was selected for this reason by the authors for the framework analysis.

Two of the researchers (JJDS, ELD) independently coded the data, they then met to establish an initial framework of the codes which emerged and generated initial overarching categories and codes. The two researchers then mapped the initial categories and codes along with the substantiated quotes from the consensus workshop discussion and sticky notes. They met and discussed the revised data codes and categories using the CFIR framework. One of the senior authors was available in case consensus was not reached by the two primary coders. However, the primary researchers were able to reach consensus on all items.

Therapist self-reported fidelity scores were collated and percentages for each session captured. An overall average percentage for all the sessions was then captured. The average scores on the Likert scale for each of the caregivers who graduated and completed the TAI and treatment satisfaction questions was also calculated.

### 2.7. Ethics

The research protocol received ethical approval from the University of Cape Town Faculty of Health Sciences Human Research Ethics Committee (HREC Ref: 787/2017; Duke University IRB: Pro00090868).

## 3. Results

### 3.1. Caregiver Participants Section

The results in Table 2 reflect the 14 caregiver participants who were enrolled in the study, evenly split between male and female. Of those, ages ranged from 36–56 years old. Only two had less than a High School graduation and seven had at least a Bachelor’s degree from a tertiary level institution. All participants were fluent in English. The 14 caregivers were parents to 7 children (in Table 2 participants 1 and 2 were caregivers for the same child, participants 3 and 4 for the next child, and so on). Eight graduated the full PCIT programme. Five of those completed the Therapeutic Attitude Inventory and the Treatment Satisfaction Questions post-intervention. Using the 1–5 Likert scale (with 5 indicating greatest satisfaction) all five participants had average scores of 4 or more on both instruments.

### 3.2. Therapist Fidelity Results

Based on the 219 recorded PCIT sessions, therapists rated sessions at an average of 97.4% integrity compared to the relevant integrity items included in the PCIT manual. Of the total sessions, 167 were rated as having 100% integrity. Only 20 sessions were scored below 90%.

### 3.3. Therapist Perceptions of Barriers and Facilitators to Implementation

Table 3 and Table 4 summarize the consensus barriers and facilitators to implementation of PCIT in South Africa as described by the therapists, and afterwards coded onto the CFIR model.

In the facilitators section, as shown in Table 3, codes were assigned to four of the six domains. Domains 5 and 6 (“Process of implementation” and “Characteristics of systems”) did not have any codes assigned. Domain 1 (“Characteristics of the intervention”) had the most assigned codes.

In the barriers section, as shown in Table 4 above, codes were assigned to four of the six domains. As in the facilitators section, domains 5 and 6 (“Process of implementation” and ‘Characteristics of systems’) did not have any codes assigned. Domain 1 (“Characteristics of the intervention”) had the most assigned codes.

### 3.4. Next Steps

As shown in Table 5 below, the therapist participants noted eight consensus next steps regarding implementation of PCIT in South Africa. In alphabetical order, these were: Adaptations; Creation of a PCIT hub; Expansion of PCIT in South Africa; Increase access; Increase accessibility; Research on PCIT in South Africa; Sustainable funding; Training (trainers and universities).

The importance of making appropriate adaptations to ensure the intervention is a good fit in the context was highlighted. Expanding PCIT in South Africa was regarded as an important priority and is linked closely to other next steps of increasing access for a broader population and increasing awareness of the intervention in caregivers and clinicians. Training of trainers was regarded as one way of achieving this aim, as was obtaining sustainable funding. Continuing with research that focused on the implementation of PCIT in South Africa was also regarded as a critical and on-going step. Finally, the creation of a “PCIT hub” that could co-ordinate these next steps and link with relevant partners was determined to be a useful method of ensuring successful, sustained implementation of PCIT in the country.

## 4. Discussion

To establish the feasibility of the intervention more comprehensively, this pre-pilot study investigated the barriers and facilitators to implementing PCIT in South Africa as described by clinicians in the process of being certified as PCIT therapists, delivering the intervention. To further examine PCIT in this setting it also explored therapist fidelity scores in delivering the intervention. These scores relate to early-stage overall feasibility in this context. In addition, it examined broad acceptability of the intervention for caregivers receiving the programme. Both groups, caregivers and therapists, indicated largely positive views of PCIT, suggesting favourable prospects of the intervention in this LMIC setting. Suggestions for next steps were also examined.

### 4.1. Caregiver Satisfaction and Acceptability

Caregiver reflections based on the TAI and therapy satisfaction questions were overwhelmingly positive. This is in line with previous literature on PCIT documenting the largely high rates of client satisfaction [31,68,69]. However, this is the first time such evidence has been documented in South Africa and, to our knowledge, in a LMIC setting. Despite it being a relatively small sample, it none-the-less indicates some evidence for acceptability and feasibility of the intervention in the country.

### 4.2. Therapist Fidelity Ratings

Relating to fidelity, therapists demonstrated high fidelity scores. Though these scores were self-reported by clinicians and so may be an over-estimate, it nonetheless, suggests that they remained largely true to the PCIT manual, delivering it as intended. Fidelity is regarded as critical to the successful implementation of evidence-based interventions into practice [70,71]. In relation to feasibility, it also reflects that PCIT can be implemented, at least within the parameters of this study, in a South African context. Despite the urgent need, there are few manualised interventions for child and adolescent mental health currently available in the country. Manualised interventions offer several strengths; they ensure that all families receive the same treatment components, they allow fidelity to be monitored in an objective manner, and allow for future scalability with integrity to the original intervention.

### 4.3. Therapist Perspectives on Facilitators and Barriers

Therapists reported both facilitators and barriers to implementation in South Africa. Consensus facilitators were most often assigned to Domain 1, ‘Characteristics of the intervention’. The strong theoretical basis and evidence-base for PCIT were described, along with the positive feedback from families and their commitment. PCIT’s self-generating evidence and its tradition of research were also regarded as facilitators. This is congruent with the literature on PCIT reporting its multiple advantages compared to alternative solutions [20,21,22,23,24,25,26,27]. In this study, therapists also described the coaching model, PCIT’s propensity to deepen the bond between caregiver and child, teaching caregiver skills while supporting them, and its long-term benefits. That it has evidence across a range of child populations was highlighted as being particularly important in South Africa. The perceived scalability and sustainability domains were new additions to the CFIR model, proposed for LMIC settings [56]. Nine separate codes were included in those two domains suggesting their relevance. As examples, the train the trainer model of PCIT, its manualisation, strong supervision ethos, and its dissemination in multiple countries were all cited.

Barriers to implementation were also predominantly assigned to the “Characteristics of the intervention” domain. The limited evidence for PCIT in South Africa and that the intervention largely focuses on a limited 2–7-year-old population were considered barriers. As South Africa is a LMIC, evidence for PCIT in this and other low-resource settings is required. For example, previous research documents that low-income families might be more prone to drop-out [72,73]. As a first step feasibility study, this research did not include a representatively wide South African population. Results highlight the importance of research with a broad range of families to better determine the impact of socio-economic status on treatment completion in this country. Perceived implementation challenges relating to scalability and sustainability included, the need for resources (e.g., one-way mirror, or microphone and speakers), the number of hours required to get a family to graduation, the time it takes to become a certified therapist, and that only therapists can be trained to deliver PCIT. In addition, the lack of a sustainable funding stream (e.g., for training more PCIT therapists) linked to the costs (e.g., of training, supervision, clinician time, etc.) of delivering the intervention was also raised. These barriers are regarded as initial impressions relating to implementation following a period of reflection by therapists who had recently received PCIT training, and indicate a need for further consideration in follow-up research.

Some reported barriers may reflect that the clinicians were being trained up as PCIT therapists during this process and so were relatively new to the intervention. For example, the reflection that it does not address caregiver psychopathology that could impact on its efficacy. While PCIT is not designed to directly address caregiver psychopathology there is some evidence that it reduces caregiver depression and stress [26,74,75]. In addition, one of PCIT’s defining features is its strong foundation based on attachment and behaviour theories. This is considered by previous literature as a positive feature of the intervention [63,76].

Previous literature on implementation in South Africa has highlighted the multicultural nature of the country [59]. For example, South Africa has eleven national languages and multiple cultural groupings. This diversity was also raised by therapist participants in this study and indicates the need for future adaptations of the programme (e.g., translation to other languages and cultural adaptions among other possible adaptations). The importance of developing accessible and acceptable interventions for a wider, though heterogenous population in South Africa is critical to considerations of implementation, and scalability. There is a precedent for adaptations of PCIT, which has been translated into multiple languages, including Spanish, German, Japanese, and German [77].

Many of the facilitators and barriers described by the participants have been reported in previous studies on PCIT [77]. While some of these (e.g., strong evidence base, high client satisfaction, cost of training, length of treatment, technology and resource required) are not exclusive to an LMIC setting, they may have more significance and present as larger barriers to implementation and future scalability and sustainability in low-resource settings.

The CFIR model proved useful in objectively stratifying the barriers and facilitators. The framework presented an opportunity to determine which of the domains were most prominent but further highlighted the importance of the newly proposed LMIC domains. Scalability and sustainability for example, are clearly salient for PCIT in South Africa. While the framework pointed to what domains were most prominently used it also highlighted, in their absentia, those domains that were not investigated. The “individuals involved in implementation”, “process of implementation”, and “characteristics of systems” domains had few codes assigned to them. While this early-stage study, necessarily prioritised other domains, it may be useful for future research to investigate these other areas.

### 4.4. Implementation and Stakeholders

Implementation continues to be a critical feature of research determining feasibility of interventions in diverse, real-world settings [49]. There is an increasing body of literature suggesting that in resource-limited contexts, there should be a greater focus, from the outset, on exploring interventions that can be scalable and sustainable [48]. The results from this study adds to that evidence and indicates that such factors are regarded as important considerations. While this study is too limited in scope to definitively indicate the clear feasibility of PCIT in South Africa, it does reflect positive early signs. The work by de Vries [11] suggests that in a setting with the dual burdens of high needs and limited resources, it is likely there will not be a ‘one size fits all solution’. His notion of a ‘suite of interventions’, that each have a good evidence base and can be delivered in ways that are appropriate for families, is compelling [11], p. 133. In relation to PCIT it is yet to be determined where it might be best placed in the South African health system. Though, with scalability as a priority, wider community provision should be a consideration, rather than isolated to tertiary level services.

The inclusion of stakeholders, as was the case in this project, is regarded as an important element of implementation research [78]. Considering the intervention beneficiaries (caregivers), and those delivering the intervention (therapists) is most likely to provide guidelines for those seeking scalable and sustainable interventions [48]. This current research has provided important first steps in determining their perspectives. As described by Chambers and Norton [79], mismatches can occur between end users and the intervention when the population and context differ from where an intervention was initially developed. Examining cultural contexts, organisational factors, accessibility, and resource limitations amongst others are important [79]. Parenting practices in particular are regarded as strongly culturally determined and should be carefully considered. Further exploration of other stakeholder perspectives in relation to PCIT will provide increased understanding of the implementation landscape. These stakeholders could include potential services delivery organisations (governmental, non-governmental, and including relevant decision makers), prospective end users (e.g., teachers, parents of older children), and potential referrers and therapists. In addition, examining current cultural beliefs and practices (e.g., on time-out, commonly used discipline approaches, and manualised interventions) may also be of benefit. While these may be longer term research goals, implementation theory should continue to inform research in this area.

### 4.5. Next Steps for PCIT in South Africa

Several consensus next steps were recommended by therapists in this study. None of those documented were considered the necessary priority, rather all were regarded as meaningful to the successful longer-term implementation of PCIT. Many could be implemented in parallel. Continued research, training, increased awareness of PCIT and sustainable funding are logical conclusions given the barriers and facilitators described. Initial funding through the Duke Global Health Institute Cape Town priority partnership enabled this pre-pilot study. Continued funding to allow the on-going development of this implementation research will be important and facilitate sustainability of the intervention. Costs, more broadly is a critical factor, particularly in resource-limited contexts. The recommendations for adaptation and expansion of PCIT in South Africa reflect both the positive view of the intervention and its potential. Since its initial development, PCIT has been adapted in multiple ways so that it can now be delivered in many settings (e.g., rural, urban, high and low resources) [80], through different means (e.g., telehealth, in-person) [40,81,82], in multiple languages (e.g., Spanish, Dutch, Japanese, etc.) [83], over different lengths (e.g., intensive PCIT) [40] and to differing populations (e.g., older children, those with autism spectrum disorder, foetal alcohol disorder, anxiety disorder) [69,84,85,86]. A growing evidence base has shown evidence for these many adaptations, and future adaptations are likely [31]. This bodes well for the possibility of adaptation of the intervention in LMIC and the multiplicity of cultures that reside in those settings. There is also some evidence that there may not be a need for significant adaptations for programmes like PCIT. A systematic review by Gardner, Mongomery and Knerr [87] on evidence-based parenting programmes indicates that contrary to common belief, extensive adaptations of such interventions did not appear necessary for successful transportation to different countries. While there is no clear, previous “roadmap” to follow, there is the precedent that with careful, considered research, adaptations can be made successfully. What adaptations should be prioritised and how best to do these should be carefully considered in future research. Engagement with relevant stakeholders is likely to provide guidance and the importance of gaining ‘buy-in’ for such adaptations from the developers of PCIT will be critical.

### 4.6. Limitations

Necessarily, as this was a first step pre-pilot into feasibility the number of participants included in the study was small. In addition, some of the caregiver participants dropped out of the intervention, this potential reporter bias may have impacted the interpretation of the acceptability scores that were only gathered for those who completed the programme. While not being representative, they nonetheless reflect the first evidence and positive support for the intervention in South Africa. It provides important guidance for next steps regarding research into PCIT in this context. Despite efforts, it was not possible to access the few caregiver participants who dropped out of the study early. This information may have provided a broader description of caregivers’ reflections on the intervention. Future and larger-scale pilot studies may allow for further exploration of caregiver perspectives.

## 5. Conclusions

There is an urgent need for evidence-based interventions in child and adolescent mental health that can be feasibly implemented in South Africa. PCIT, in this early-stage, pre-pilot data shows promise as an acceptable and accessible intervention in South Africa. Therapist views indicated many facilitators for implementation, including strong evidence base, manualisation, and training model. Consideration for barriers, particularly relating to sustainability and scalability were highlighted. Largely positive views relating to acceptability from caregiver participants also indicated the promise of PCIT as an intervention in this context. Pilot data on the efficacy of the treatment for participating families are needed as a next step. While initial results are positive, on-going research of how implementation factors can contribute to the longer-term successful dissemination of PCIT in complex and heterogeneous LMIC settings is required.

## Figures and Tables

**Figure 1 ijerph-19-04450-f001:**
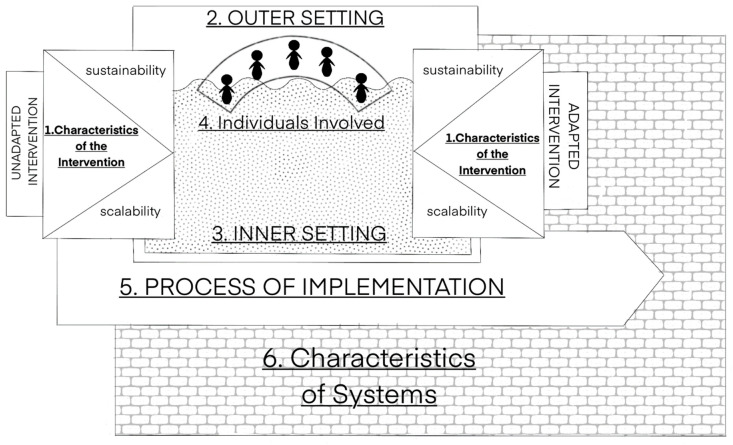
Representation of the Consolidated Framework for Implementation Research (CFIR) adapted for LMIC settings, as proposed by Means et al. (2020).

**Figure 2 ijerph-19-04450-f002:**
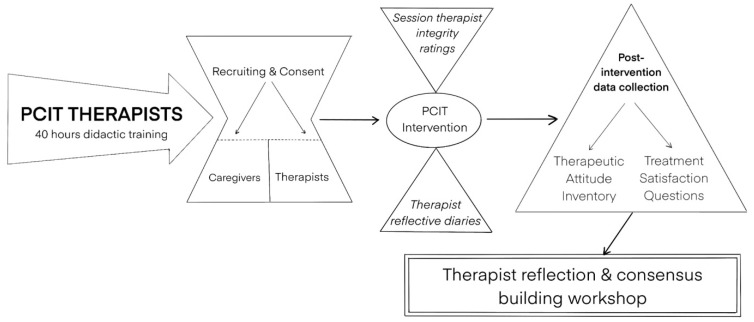
Graphic representation of the research process.

**Table 1 ijerph-19-04450-t001:** Key implementation terms (adapted from Kumm et al. [53]).

Construct	Definition	Reference
Overall Feasibility	The extent to which a new intervention can be used successfully within a given setting, including elements of implementation (e.g., acceptability, fidelity).	Karsh [54]
Acceptability	The perceived fit, relevance, or compatibility of an intervention to a particular user, provider, community or setting	Proctor et al. [51]
Fidelity	The extent to which an intervention was implemented as it was prescribed in the original protocol or as it was intended by the developers of the programme	Proctor et al. [51]
Scalability	The ability of an intervention (shown to be efficacious on a small scale and/or under controlled conditions) to be expanded to reach a greater proportion of the eligible population, while retaining effectiveness in real world conditions	Aarons et al. [55]

**Table 2 ijerph-19-04450-t002:** Caregiver demographics and PCIT graduation results.

Caregiver	Age	Gender	Self-Identified Race/Ethnicity	Education	Graduation/Completion
Participant 1	42	Female	White	Bachelors	Y
Participant 2	38	Male	White	Diploma	Y
Participant 3	39	Female	Coloured	Less than high school graduation	Y
Participant 4	39	Male	Coloured	Less than high school graduation	Y
Participant 5	41	Female	White	Bachelors	Y
Participant 6	45	Male	White	Bachelors	N
Participant 7	38	Female	Coloured	Diploma	Y
Participant 8	40	Male	Coloured	Diploma	Y
Participant 9	36	Female	Black	Bachelors	N
Participant 10	56	Male	White	High school	N
Participant 11	36	Female	White	Bachelors	N
Participant 12	40	Male	White	Bachelors	N
Participant 13	37	Female	White	Bachelors	Y
Participant 14	40	Male	White	Diploma	N

Note: In the South African Census Data race/ethnicity is self-declared in four main categories: Black, Indian, Coloured and White. For this reason, the same system was used here.

**Table 3 ijerph-19-04450-t003:** Facilitators to implementation.

Domains and Constructs	Therapist Consensus Reflections
**Domain 1: Characteristics of the Intervention (relating to the quality and features of the intervention)**	
Evidence Strength and Quality: Perception of the quality and validity of evidence supporting the belief that the intervention will have desired outcomes	Strong theoretical basis Data collection—easy to build evidence Strong tradition of research Good evidence base Research has facilitated PCIT in South Africa Feedback from parents has been positive Good technical resources Family commitment Feedback from parents has been positive
Relative Advantage: Perception of the advantage of implementing the intervention versus an alternative solution	Attending supportive training programme Innately rewarding, positivist, encouraging Develops language of children Facilitates parents being able to listen to their children Long term benefits for families Deepens bond between parent and child Coaching model—parents are learning skills, supportive for parents More than one caregiver strengthens their relationship with their child Teaches parents skills that facilitates their relationships with their child Can work in conjunction with other interventions Focuses on the relational—which is everything Clear beginning, middle and end PCIT can be done with a range of children/conditions—ASD, ODD, anxiety, etc. Gives psychological resources to parents
Adaptability: Degree to which an intervention can be tailored to meet the needs of an organization	PCIT can be adapted (as evidenced by multiple adaptations to the initial and core intervention)
Complexity: Perceived difficulty of implementation	Clear model
Perceived scalability	Can work as a specialist service in a tertiary service Train the trainer model Manualized intervention Fidelity—therapists can be measured Has been disseminated in many countries and low-income contexts Can be done remotely (e.g., via telehealth/online) PCIT International has a history of encouraging research and practice relating implementation and dissemination
Perceived sustainability	Supportive supervision and peer supervision Supervision that facilitates learning the model
**Domain 2: Outer Setting (referring to the economic, political and social contexts where the tertiary level hospital resides)**	
Patient Needs and Resources: Extent to which patient needs are accurately known and prioritized by the organization	Significant need for intervention in South Africa Can change the pathway for those who might otherwise progress to conduct disorder
Cosmopolitanism: Level of connectedness and networks with other organizations	Support from DCAP, NDF, Duke University, Nussbaum Foundation Support from US partners
**Domain 3: Inner Setting (refers to the structural, political and cultural contexts where the implementation will take place, e.g., the hospital, department group of people)**	No direct data
**Domain 4: Individuals involved in implementation (referring to those involved in implementing the intervention)**	
Knowledge and Beliefs about Intervention: Individual staff knowledge and attitude towards the intervention	Clinician motivation High clinician commitment Excellent training and enthusiastic trainee PCIT therapists
**Domain 5: Process of implementation (referring, though not limited to, the planning around implementation and execution of that plan)**	No direct data
**Domain 6: Characteristics of systems (referring to the relationship between key systems characteristics and implementation)**	No direct data

**Table 4 ijerph-19-04450-t004:** Barriers to implementation.

Domains and Constructs	Therapist Consensus Reflections
**Domain 1 Characteristics of the Intervention (relating to the quality and features of the intervention)**	
Evidence Strength and Quality: Perception of the quality and validity of evidence supporting the belief that the intervention will have desired outcomes	Evidence to date only from specialist CAMH team—need for piloting/implementation in less specialised community and rural contexts Cultural acceptability in a broader range of South African populations still required given sample participants are not representative of the country Further evidence needed it is acceptable for a larger cohort of South African parents
Relative Advantage: Perception of the advantage of implementing the intervention versus an alternative solution	Concern that the focus of PCIT is largely on behaviour and attachment and not more fully aware of caregiver psychological issues that may prevent them from implementing effective parenting strategies (what an alternative intervention or programme would be that does that, was not described) Very specialised intervention Only a few can access it Only focuses on 2–7-year-old cohort
Adaptability: Degree to which an intervention can be tailored to meet the needs of an organization	Adaptations likely required to ensure it is acceptable and scalable in a South African context, unclear the extent of adaptations required Language barrier—only available in English (of the 11 national languages in South Africa), translation needed to allow access to wider population
Complexity: Perceived difficulty of implementation	Low-income families may have to prioritise work over treatment Family instability can impact on caregiver uptake of the intervention Parental mental health Limited parental psychological resources can impact on their capacity to engage with the intervention For different reasons, families not always displaying sustained effort
Cost: Cost of the intervention and costs associated with implementing the intervention	Dependence on technology—particularly with electricity load shedding or blackouts (a relatively regular occurrence in South Africa currently) Financial constraints Cost of training Costly infrastructure Cost of technology
Perceived scalability	Many resources required to implement the intervention which the majority of health care settings in South Africa would not have access to (e.g., one-way mirrors, headsets, microphones) Caregivers often needed extra support in addition to the 1 h of standard PCIT a week Time—the number of man hours required to get a family to graduation Time it takes to become certified Only therapists can be trained—cannot be widely disseminated
Perceived sustainability	Current lack of sustainable funding stream for PCIT training and clinical services in South Africa High commitment required from families (time and finances) Concern about length of treatment unsustainable for families with limited resources
**Domain 2: Outer Setting (referring to the economic, political and social contexts where the tertiary level hospital resides)**	
Cosmopolitanism: Level of connectedness and networks with other organizations	To the therapist knowledge, only limited presence of PCIT in LMICs—therefore no roadmap Aside from regular contact and supervision with Global Trainer, limited other connections with a larger PCIT community No structural hub coordinating PCIT in South Africa No government buy in
**Domain 3: Inner Setting (refers to the structural, political and cultural contexts where the implementation will take place, e.g., the hospital, department, group of people)**	
Implementation climate: Relative priority of implementing the current intervention versus other competing priorities	Lack of work balance—difficult to create space for PCIT in work schedule Lack of management buy-in
Readiness for Implementation: Access to resources, knowledge, and information about the intervention	Model is unknown by clinicians—therefore limited referrals
**Domain 4: Individuals involved in implementation (referring to those involved in implementing the intervention)**	
Self-efficacy: An individual’s belief in their capabilities to execute the implementation	Given the therapists were bring trained during the course of the study, they were relatively inexperienced in the model
**Domain 5: Process of implementation (referring, though not limited to, the planning around implementation and execution of that plan)**	No direct data
**Domain 6: Characteristics of systems (referring to the relationship between key systems characteristics and implementation)**	No direct data

**Table 5 ijerph-19-04450-t005:** Consensus next steps as determined by therapist participants.

Adaptations
Creation of a “PCIT hub”
Expansion of PCIT in South Africa
Increase access
Increase awareness
Research
Sustainable funding
Training (trainers and universities)

## Data Availability

The data presented in this study are available on request from the corresponding author. The data are not publicly available due to privacy reasons.

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
