# Peer review of "Bringing Parent–Child Interaction Therapy to South Africa: Barriers and Facilitators and Overall Feasibility—First Steps to Implementation"

_ijerph, 2022, doi:10.3390/ijerph19084450_

Round 1
Reviewer 1 Report
publish paper as is.
This manuscript is a resubmission of an earlier submission. The following is a list of the peer review reports and author responses from that submission.
Round 1
Reviewer 1 Report
Dear Authors,
Overall, the article is well written, and presents information relevant to the area of study. Congratulations.
I believe the article is suitable for publication with minor corrections.
First of all, I recommend a formatting correction. Especially the tables. But also font, etc..
In addition to this, in the methodology I would add a diagram that would allow to follow the flow of the research. In the same way, I would include and develop the information related to the "Consolidated Framework for Implementation Research (CFIR)" in this section, since it corresponds to a methodological strategy.
Best wishes,
Reviewer
Reviewer 2 Report
Thank you for the opportunity to review this interesting paper describing the introduction of PCIT to South Africa.
More significant points
The paper tries to cover a lot of ground for a relatively small (but interesting) study. And at times the authors make claims that are quite a stretch given the scope of the study. The study aims to explore barriers and facilitators to implementation, “overall feasibility”, “therapist integrity” / fidelity (line 191), “acceptability” to clinicians (line 234), and acceptability to families (line 255) / treatment satisfaction (line 190). It also claims to provide information on “longer-term sustainability” (line 361). Many of these terms have specific meanings within the implementation science literature (see Proctor et al. (2011)).
I have the following suggestions to strengthen the paper:
- Particularly given that the authors are planning further research to extend this work, it seems important to accurately describe the place that this study occupies. “Pre-pilot study” is not a commonly used term, but I can see why the authors have used it here. However it would be useful to clarify the aims section, and revise the abstract accordingly (specifically, clarifying the “overall feasibility” mentioned in the abstract and elsewhere, which needs to be more specific). For example, is the primary aim of this study to understand barriers and facilitators, and secondary aim to report preliminary data on acceptability to clinicians and families?
- Carefully review the use of terms such as feasibility, acceptability, fidelity, sustainability throughout the paper, perhaps with reference to Proctor et al. (2011)’s definitions, and ensure these terms are used validly and consistently.
- Add in additional caveats in the Discussion section to aid the reader to interpret the findings. For example;
- Fidelity (also referred to as integrity in the paper) was only self-reported by clinicians and may be an over-estimate.
- The TAI was only completed by 5 of the 8 caregivers who completed the programme? This ought to be noted as a limitation in the interpretation of acceptability to families – those who complete the programme are more likely to like the programme.
- The study included a small number of (presumably) highly motivated clinicians, who may have been grateful for the opportunity to receive training, and (presumably) were provided with management support, perhaps by way of dedicated time for PCIT or access to equipment. These considerations relate to why this study can’t really draw conclusions around “longer-term sustainability” (line 361), for example.
Minor points
The authors may be aware that there is a body of research around the “transportability” of manualized parent training programmes from their country of origin, and the Introduction or Discussion sections would be strengthened with (brief) reference to this literature. It isn’t specific to PCIT, but implies that – contrary to expectations – relatively few adaptations may be required, even in LMIC. For example, Gardner, Montgomery & Knerr (2016)’s review.
Lines 112-119 detail countries outside of the USA where PCIT has been disseminated, and could usefully include Australasia where PCIT was introduced many years ago. Specifically, Australia (e.g., Dr Jane Kohlhoff’s (nee Phillips) community-based work) and New Zealand (e.g., Dr Melanie Woodfield’s implementation work).
It would be useful to have more information around the “purposive” selection that occurred – it seems that this wasn’t really selection as such, but included all clinicians who received the initial 40-hour training, and all families who received PCIT? This could also be described as a convenience sample. (This doesn’t necessarily detract from the usefulness of the study, but is important to clarify.)
Line 231 is unclear – was the workshop facilitator also a clinician and participant? Or did the workshop facilitator, along with a clinician, and a participant, generate the list? What relationship did the researcher(s) have to these participants? This becomes more relevant in a qualitative study, as it influences the interpretive lens applied.
The “Data Analysis” section is well-written and useful.
Thanks again for the opportunity to review this interesting work, and I wish the authors well for the next steps on their journey of exploring PCIT’s implementation in South Africa.
Reviewer 3 Report
Excellent paper. The paper doesn't provide observation data on children's behavioral disorders--use examples of disruptive behavior not jargon-- and, most importantly, I missed the culture context of those children, parents handling of children's disorders, and native languages spoken at home. The word language was used seven times. What languages do children speak? Were there language barriers? How are children's behavioral disorders handled within their homes by their parents? Authors should explain how an American university became involved in the behavior of South African children. I might have missed it, but do the U.S. authors find their behavior, their intrusion, into families in South Africa troubling? The paper is over-cited; too many citations interfere with reading.
